# Multiplets in scRNA-seq data: Extent of the problem and efficacy of methods for removal

**Dimitris Ttoouli**[ID]*, **Daniel Hoffmann**[ID]

Bioinformatics and Computational Biophysics, Faculty of Biology, University of Duisburg-Essen, Essen, Germany

\* dimitris.ttoouli@uni-due.de

## Abstract

Multiplets—droplets that capture more than one cell—are a known artefact in droplet-based single-cell RNA sequencing (scRNA-seq), yet their prevalence and impact remain underestimated. In this study, we assess the frequency of multiplets across diverse publicly available datasets and evaluate how well commonly used detection tools are able to identify them. Using cell hashing data to determine a lower bound of the true multiplet rate, we demonstrate that commonly used heuristic estimations systematically underestimate multiplet rates, and that existing tools—despite optimized parameters—detect only a small subset of cell-hashing multiplets. We further refine a Poisson-based model to estimate the true multiplet rate, revealing that actual rates can exceed heuristic predictions by more than twofold. Downstream analyses are significantly affected by multiplets: they are not confined to isolated clusters but are distributed throughout the transcriptional landscape, where they distort clustering and cell type annotation. In differential gene expression analysis, multiplets inflated artefactual signals while expected cell-type markers remained stable, leading to shifts in effect sizes and partial loss of significant genes despite high overall fold-change correlation. Using both quantitative and qualitative approaches, we visualize these effects and show that cell-hashing-informed multiplet removal eliminates artefactual clusters and improves annotation clarity, whereas computationally detected multiplets fail to fully remove artefacts in the most common experimental contexts. Our findings confirm that multiplet contamination remains a pervasive and under-addressed issue in scRNA-seq analysis. Since most datasets lack multiplexing, researchers must often rely on heuristics and limited tools, leaving many multiplets unidentified. We advocate for more robust multiplet-detection strategies, including multimodal validation, to ensure more accurate and interpretable scRNA-seq results.

## Introduction

Single-cell RNA sequencing (scRNA-seq) was named Method of the Year in 2013 by Nature Methods and has been rapidly advancing ever since [1]. This manuscript will focus on one of

**Data availability statement:** All datasets required to reproduce the analyses reported in this manuscript are available at Zenodo under DOI: https://doi.org/10.5281/zenodo.17315220.

**Funding:** This study was funded in part by grant HO 1582/12 from Deutsche Forschungsgemeinschaft. The funders had no role in study design, data collection and analysis, decision to publish, or preparation of the manuscript.

the most widely used scRNAseq technologies at the moment, which involves micro-fluidic droplets that ideally encapsulate single cells with barcoded gel beads [2–4]. The gel bead provides not only a unique barcode for each droplet, but also a unique molecular identifier (UMI) for each transcript captured within each droplet, so that any PCR amplification bias can be overcome and only the number of captured transcripts for each droplet is counted [5]. After the amplification of captured transcripts, the library can be sequenced with a high-throughput sequencing (HT-seq) machine.

An artefact that can arise in any single-cell technology available today, but is especially prominent in droplet-based methods, is if two or more cells, for biological or technical reasons, end up in the same droplet—a so-called multiplet. Multiplets are problematic for the analysis because a collection of cells is mistakenly interpreted as a single cell. There are many factors that are known to influence the fraction of multiplets in any given single-cell experiment e.g. cell interactions and number of cells loaded in a single experiment. Since the true number of multiplets cannot be experimentally inferred by simple means, the fraction of multiplets is usually an estimate, which has been reported in literature to range from as low as 5% to as high as 40% [6,7]. By extension, the true number of singlets, i.e. individual single cells captured inside individual droplets, is also usually not known.

Multiplets have been shown to affect the analysis steps that come after sequencing (downstream tasks) by cluttering the signal of true single cells and adding to the noise that carries over and accumulates with each next analysis step [8]. It has also been demonstrated that multiplets can be mistaken for new cell types [9]. Consequently, the removal of multiplets improves downstream task quality and interpretation, and for this reason, multiple tools have been developed for identifying multiplets in single-cell datasets [10–15]. A major limitation of such tools is that they can only identify multiplets between transcriptionally different cells (heterotypic multiplets) and cannot distinguish multiplets between cells of the same transcriptional activity or cells within the spectrum of differentiation of a single cell type (homotypic multiplets). Assessing cells from the same tissue or even the same major cell type, however, is a commonly performed experiment, e.g. experiments on brain cells or experiments on isolated T cells. Some methods exist that facilitate multiplet detection using more than the transcriptome [16,17]. These methods, however, are limited to single-cell experiments that investigate other modalities such as surface protein expression and/or immune receptor expression on top of the transcriptome, and their usage in peer-reviewed literature has not yet been extensive.

The performance, efficacy and accuracy of multiplet-detection tools have been assessed in a benchmarking study by Xi and Li [7]. However, because singlets cannot be easily identified with the current technology, whether experimentally or computationally, there are no ground-truth labels for published datasets. Yet, there is a way to experimentally tag two or more samples with either different antibodies or different lipid-tags (cell hashing); in this way multiplets formed between samples can be identified [18,19]. Still, multiplets from the same sample will remain unknown. Previous studies have used tagged multiplets as quasi-ground-truth, even though they are only a lower bound on the number of multiplets within a dataset, and consequently, the extent of their influence on downstream tasks is also a lower bound.

## Materials and methods

The benchmarking study [7] evaluated all available multiplet-detection tools at the time, including scds (with methods bcds, cxds, and hybrid) [10], DoubletCells [11], DoubletDetection [12], DoubletFinder [13], scDblFinder [14], Scrublet [15], and Solo [6]. Today there are a few more multiplet-detection tools, including various machine learning tools, which claim

state-of-the-art performance. A bibliographic search reveals that the overwhelming majority of new single-cell research uses either DoubletFinder or Scrublet to remove multiplets (Fig 1), which were the tools identified by [7] as the best performing in their benchmark study. For this reason, this manuscript uses code from [7], adapted to work with the latest versions of the multiplet-detection tools included in it. However, it must be clearly stated that the purpose here is not to compare the efficacy, accuracy or scalability of the tools, but rather to show that the majority of analyzed single-cell transcriptomics datasets in literature today probably include unidentified multiplets, which implies consequences for the interpretation of the analyses' results. Newer tools were not included in the present work, since this is not a benchmarking study and these tools are not frequently cited in literature. The approach followed here was to use the multiplet-detection tools to generate multiplet scores for each droplet, using the most optimal parameters for each tool in each dataset. As in [7], instead of using droplet label predictions (i.e. singlet or multiplet) as tool outputs, the scores that the tools assign to droplets are ranked from highest (tool is confident this is a multiplet) to lowest (tool is confident this is a singlet) and the number of top-ranked droplets that maximizes the area under the precision-recall curve (AUPRC) is selected. Even though cell-hashing multiplets were used as the quasi-ground-truth to compute the PRC, we remind the reader that these are a lower bound of the actual multiplet count and are only used for this purpose for a lack of better alternatives. Solo, instead of a score that correlates with the likelihood that a droplet is a multiplet, outputs a classification label and a confidence score for the classification, and thus cannot be directly compared with the other tools using the above-described methodology, so it had to also be excluded.

The order of operations in the pipeline used for this study is shown in Fig 2. The following paragraphs were written in the same order as the steps of the pipeline, so that the reader can reference each step in the flowchart along with the text that follows.

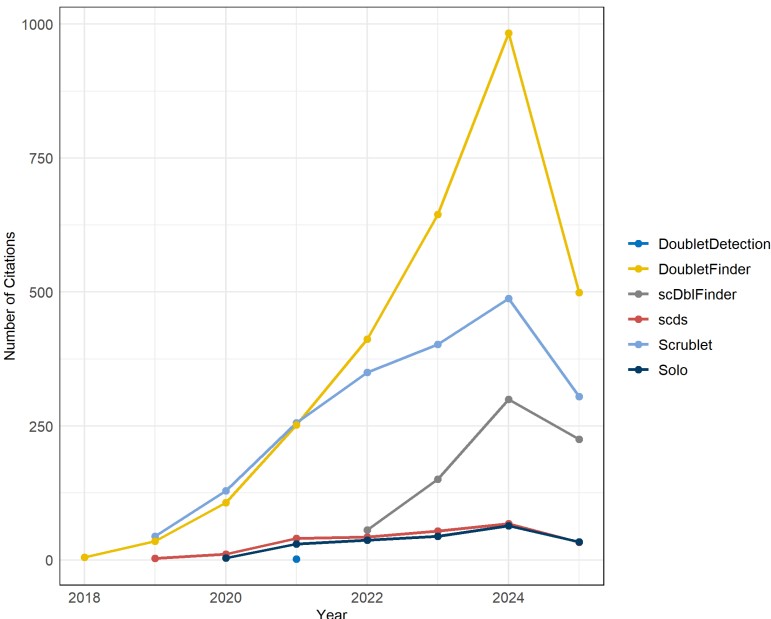

**Fig 1. Citations over time for multiplet-detection tools.** Number of citations per year for the multiplet-detection tools included in this study. DoubletFinder and Scrublet are by far the most popular choice for newer single-cell research.

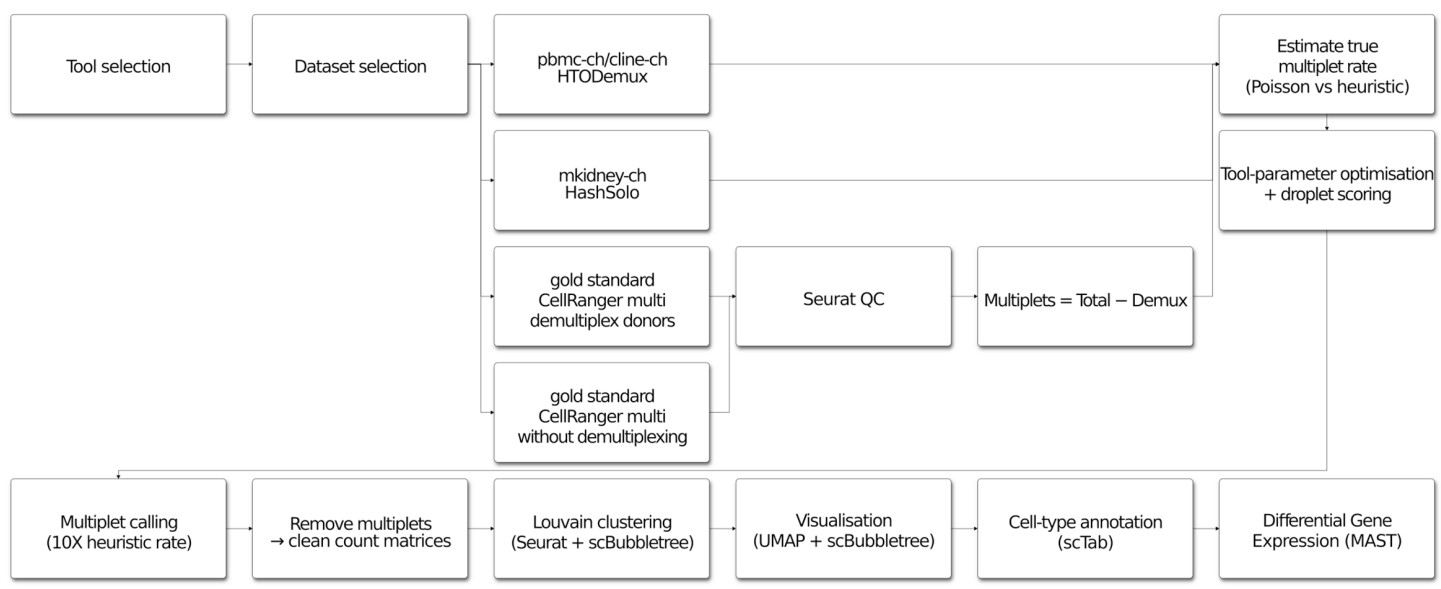

**Fig 2. Pipeline overview.** Overview of the pipeline used in this study for analyzing multiplets in scRNA-seq data.

## Datasets

To investigate the impact of multiplets on scRNAseq datasets we used four publicly-available, droplet-based datasets, produced using 10X Genomics' technology, with cells from cell lines, mouse cells and human cells, briefly described here:

pbmc-ch, from the Gene Expression Omnibus (GEO) accession number GSE108313 [18], was produced using human Peripheral Blood Mononuclear Cells (PBMCs) from eight donors with 10X Genomics 3' version 2 chemistry and contains 15,272 droplets, of which 2,545 are identified by cell hashing as multiplets, which corresponds to 16.66% multiplet rate.

cline-ch, also from the GEO accession number GSE108313 [18], was produced using four human cell lines, HEK293T, K562, KG1, and THP1, also with 10X Genomics 3' version 2 chemistry and contains 7,954 droplets, of which 1,465 are identified by cell hashing as multiplets, which corresponds to 18.42% multiplet rate.

mkidney, from the GEO accession number GSE140262 [6], was produced using two samples of dissociated mouse kidney cells with 10X Genomics 3' version 3 chemistry and contains 21,179 droplets, of which 7,901 are multiplets, which corresponds to 37.31% multiplet rate.

The 'gold standard' dataset, was produced by 10X Genomics to illustrate a workaround analysis pipeline for their Immune Profiling 5' sequencing product (simultaneous whole transcriptome, panel of surface proteins and paired TCR/BCR sequencing) using their own analysis software, CellRanger [20]. The dataset was produced using PBMCs from two healthy human donors (donors 1 and 2), Bone Marrow Mononuclear Cells (donor 3) and PBMCs (donor 4) from two Acute Lymphocytic Leukemia (ALL) patients. To avoid unexpected interactions stemming from abnormal gene expression in cancer cells, all droplets (including cell-hashing multiplets) associated with donors 3 and 4 with ALL were excluded from the analysis. Given the presence of human cancer cell lines in the cline-ch dataset, we deemed it more appropriate for this dataset and the purposes of this manuscript to focus exclusively on healthy donors. After the removal of ALL patients, the dataset contains 27,504 droplets, of which 7,186 are multiplets, which corresponds to 26.13% multiplet rate. It must be stressed

here that the gold standard dataset is unique, in the sense that to the authors' best knowledge there are no other Immune Profiling datasets combined with cell hashing, a feature that can be used to shed further light on the problem of multiplets using information other than the transcriptome (hence the nickname gold standard), which is, however, outside the scope of this study.

### Dataset preprocessing

The datasets pbmc-ch, cline-ch and mkidney-ch were downloaded already preprocessed from the online research repository Zenodo, as they were made publicly available by [7], where the authors also included labels for singlet or multiplet for each droplet [21]. Briefly, for pbmc-ch CD45 and for cline-ch CD29 and CD45 were used as the hashing antibodies and both datasets were demultiplexed using Seurat's HTODemux function [18,22]. For mkidney-ch, cholesterol-modified oligos were used for cell hashing, and the dataset was demultiplexed using a custom-made probabilistic demultiplexing method, HashSolo, developed for this dataset by the same authors that produced it [6].

The gold standard dataset was processed according to 10X Genomics' guide for demultiplexing 5' Immune Profiling datasets with cell hashing [23]. In a nutshell, the multiplexed libraries were processed once with CellRanger multi to demultiplex each library (gene expression, surface protein expression, TCR and BCR) to the four different donors. At this point the cell-hashing multiplets were removed by CellRanger. Then, the two healthy donors' libraries were converted from the BAM output of CellRanger back to FASTQ using bamtofastq (which is bundled with CellRanger) and then processed the healthy donors' libraries separately with CellRanger multi once more to complete the pipeline. The multiplexed dataset was independently run with CellRanger multi without demultiplexing, to get the output that includes the cell-hashing multiplets. In the demultiplexed version of the dataset CellRanger removed the multiplets so that only droplets that are considered singlets by cell hashing from the two healthy donors remain. The multiplexed version of the dataset contains both singlets and multiplets from all four donors. To create the final version of the dataset, the droplets of the demultiplexed version and the cell-hashing multiplets with tags of both healthy donors from the multiplexed version were combined using Seurat. The minimum number of genes was set to 500 and the minimum number of cells expressing a single gene for it to be included in the dataset was set to 3, to filter out broken cells and cell-free droplets.

### Approximation of the true number of multiplets

The encapsulation of cells in micro-fluidic droplets can be thought of as a Poisson process with parameter $\lambda$ representing the average number of cells loaded per droplet (including empty droplets) [24]. The probability that a droplet contains $k$ cells is given by

$$P(k) = \frac{e^{-\lambda}\lambda^k}{k!}. \tag{1}$$

It follows then that the probability of a droplet being empty is $P(0) = e^{-\lambda}$ and the probability of a droplet containing exactly 1 cell is $P(1) = e^{-\lambda}\lambda$. The fraction of non-empty droplets out of all the droplets is $1 - P(0) = 1 - e^{-\lambda}$, and among these, the fraction of droplets containing exactly one cell (i.e. singlets) is

$$\frac{P(1)}{1 - P(0)} = \frac{e^{-\lambda}\lambda}{1 - e^{-\lambda}}. \tag{2}$$

Hence, the fraction of non-empty droplets that are multiplets, i.e. the true multiplet rate of the dataset, is

$$f_{\text{mult}} = 1 - \frac{\lambda e^{-\lambda}}{1 - e^{-\lambda}}. \tag{3}$$

In experiments where cells from $D$ donors or samples (henceforth summarily called "samples", for simplicity), are pooled using cell hashing, each cell is labeled with a sample-specific tag. Because only multiplets containing cells from different samples can be identified, the observed inter-sample multiplet rate, $f_{\text{obs}}$, underestimates the true multiplet rate. A droplet containing $k$ cells, assuming equal sample representation and random assignment, will consist entirely of cells from a single sample with probability

$$\frac{1}{D^{k-1}}, \tag{4}$$

so that the probability that a multiplet contains cells from different samples is given by:

$$1 - \frac{1}{D^{k-1}}. \tag{5}$$

When this probability is weighted by the Poisson probability of all multiplets ($k \geq 2$) and averaged over all multiplets, we get the fraction of multiplets that contain cells from different samples, i.e. the theoretical inter-sample multiplet fraction:

$$f_{\text{diff}} = \frac{\sum_{k\geq 2} P(k) \left(1 - \frac{1}{D^{k-1}}\right)}{\sum_{k\geq 2} P(k)}. \tag{6}$$

Accordingly, the observed inter-sample multiplet rate $f_{\text{obs}}$ can be expressed as the product of the true multiplet rate and the theoretical inter-sample multiplet fraction:

$$f_{\text{obs}} = f_{\text{mult}} f_{\text{diff}}. \tag{7}$$

Because $f_{\text{obs}}$ is directly measured from the experiment, we can numerically solve

$$f_{\text{mult}}(\lambda) f_{\text{diff}}(\lambda, D) = f_{\text{obs}} \tag{8}$$

for $\lambda$. Once $\lambda$ is estimated, we have an effective droplet occupancy parameter that describes the distribution of the number of cells in non-empty droplets. Finally, because only inter-sample multiplets are identifiable by cell hashing, the true multiplet rate among non-empty droplets $f_{\text{mult}}$ can be directly calculated by

$$f_{\text{mult}} = \frac{f_{\text{obs}}}{f_{\text{diff}}}. \tag{9}$$

This framework thus enables us to estimate the true multiplet rate from the observed inter-sample multiplet rate by accounting for the Poisson statistics of cell loading into droplets and the effect of sample multiplexing.

## Parameter optimization and multiplet scoring

In this manuscript, the parameters of tools were optimized exactly as in [7] by iterating through a range of parameter value combinations and selecting the values that maximize the area under the AUPRC for multiplet scores, using cell-hashing multiplets as ground-truth to check against [7]. The range of values explored were chosen based on the tools' own recommended values for their parameters. After the optimal parameter values for each method and dataset were selected, the tools were run with those parameters to score the droplets of each dataset. The accuracy of each tool against cell-hashing multiplets was then assessed on the basis of the area under the receiver operating characteristic curve (AUROC).

## Multiplet calling and removal

The multiplet droplets were identified for each dataset using once more code from [7]. The approach here is that multiplets are simply called by assuming a rate of multiplets and selecting the number of top-scored droplets that correspond to that rate for the number of droplets in the respective dataset, e.g. 10% out of 12,000 non-empty droplets. In this study we used the theoretical rate of multiplets that can be approximated using a simple rule provided by 10X Genomics for multiplet calling, which consistently underestimates the number of multiplets (see Results). This happens because 10X Genomics' suggests a heuristic to estimate the non-observed number of multiplets using the observed number of non-empty droplets retrieved at the end of experiment, which does not cover all the factors that influence multiplet formation. One could correctly argue that using the rate of multiplets determined by cell hashing can significantly improve this estimation, even if cell-hashing multiplets are a lower bound of the true number of multiplets. Most users, however, don't multiplex their samples and prefer the more straightforward and cheaper experiment, which does not retain any information about the number of multiplets. This means that, in most cases, a researcher only has 10X Genomics' estimation to go by and this is why this theoretical rate of multiplets was used instead of the observed experimental rate of multiplets.

Once multiplets were called, different versions of each dataset's count matrix were created where the multiplets identified by either a tool or cell hashing were removed to assess the impact on downstream tasks. The removal was done using code from [7].

## Clustering

Each version of each dataset's count matrix was clustered using the Louvain algorithm [25]. Each dataset was first converted into a Seurat object, then normalized, scaled, and the top 2000 most variable genes were identified [22]. We followed this with a principal component analysis (PCA) on these genes and a k-nearest neighbor graph was then constructed based on the top ten principal components of PCA and the nearest neighbors of each droplet were identified. Finally, Louvain clustering was applied using Seurat's implementation, using default parameters.

## Clustering quality assessment

We evaluated clustering quality using the clusterCrit package [26]. Three metrics were selected: the Calinski–Harabasz Index (higher is better), which rewards global separation and a higher number of compact clusters; the Davies–Bouldin Index (lower is better), which rewards compact clusters that are well-separated from their nearest neighbors; and the Silhouette value (closer to 1 is better), which rewards droplets being closer to their own cluster than to other clusters. To account for differences in dataset size after multiplet removal,

we performed a stratified subsampling prior to calculating clustering metrics. For each dataset version, we randomly subsampled droplets 100 times, without replacement, to a fixed number equal to the largest rounded count below the droplet count of the smallest dataset version. Subsampling was stratified by cluster membership to preserve relative cluster composition, and the metrics were calculated on the existing cluster assignments without reclustering. Finally, we derived an average rank across methods by ranking the median performance of each method separately within each dataset and metric combination according to metric direction (descending for Silhouette value and Calinski–Harabasz, ascending for Davies–Bouldin). To account for consistency, we penalized methods with wider interquartile ranges, by ranking the interquartile ranges within each dataset and metric combination as well, and added these ranks to the performance ranks with a small weight (0.1). We then averaged the ranks across all dataset and metric combinations and also across all datasets only for the Silhouette value.

## Visualization

Uniform Manifold Approximation and Projection (UMAP) was used to visualize the clustering results and the density of multiplets for each dataset and method. Seurat's `UMAPPlot` function and a customized version adapted for density plotting were used [22]. To avoid problems due to severe overplotting, scBubbletree was used to create a more quantitative way of presenting the same results [27]. Bubble trees for the unprocessed and cell-hashing versions of each dataset were created by scBubbletree's `get_bubbletree_graph` function, using the same parameters and inputs as in the Louvain clustering step in Seurat, over 200 bootstrapping iterations. Vertically and horizontally integrated heatmap tiles of the distribution of multiplets within datasets and within each dataset's clusters respectively, were created with scBubbletree's `get_cat_tiles` function using default parameters.

## Cell type annotation

ScTab was used to annotate the clusters of each dataset [28]. The tool leverages a pre-trained deep learning model, which was specifically engineered for tabular data, and uses a novel approach to augment its large training data corpus of about 22.2 million cells to predict cell types. The advantage of scTab over other tools is not only its reported accuracy and efficient scaling over large datasets, but also its ability to predict cell types for individual droplets, instead of entire clusters. In this way, we can see the composition of clusters mixed from different cell types. The tool was ran with the pre-trained model state, model architecture and hyperparameters that are provided and demonstrated in the inference tutorial of the tool's documentation [29]. The gene naming scheme in the datasets in this work and the one used for the training set of scTab are different and a translation between them was necessary in order to use scTab. For the translation between gene naming schemes we used the package biomaRt [30]. Since scTab annotates each individual droplet and not an entire cluster, the most numerous annotation in each cluster was assigned as the cluster's cell type annotation.

## Differential gene expression

Differential gene expression (DGE) between cluster pairs was performed using the MAST package [31]. The gold standard dataset was imported retaining cell-hashing multiplet, clustering, and cell-type annotation information. To ensure robust testing, only genes expressed in at least 5% of droplets were considered for DGE. DGE comparisons were carried out between

pairs of clusters from the "No processing" version of the dataset. For each comparison, analyses were run twice: (1) using all droplets from the relevant clusters, (2) excluding multiplets, without re-running the entire preprocessing pipeline. DGE was then tested using MAST's `zlm` function, using method "bayesglm", cluster membership as the predictor, and the first cluster of the pair as reference. Wald tests were performed for the coefficient of the second cluster, and p-values were adjusted for multiple testing using the Benjamini–Hochberg procedure. Genes with false discovery rate (FDR) less than 0.05 and absolute $\log_2$ fold change over 0.5 were considered significant. To quantitatively compare the effect of multiplet removal on DGE results, the Jaccard index of the overlap between significant gene sets, and the Spearman correlation of $\log_2$ fold changes were computed on the union of significant genes across both analysis runs (with and without cell-hashing multiplets). Volcano plots were generated to visualize significant genes, with the ten most strongly differentially expressed genes annotated. Additionally, a $\log_2$ fold change comparison plot was produced to compare effect sizes between the two analysis runs.

## Results

### Multiplet fraction is underestimated

Accurately determining multiplet fractions is a fundamental yet challenging issue in single-cell analysis. 10X Genomics provides a heuristic for estimating multiplet frequency based on the number of non-empty droplets at experiment completion. We assessed how closely this heuristic aligns with actual multiplet rates identified by cell hashing, using datasets derived from diverse sources, including cell lines, mouse tissues, and human peripheral blood mononuclear cells (PBMCs). These datasets also involved different sequencing chemistries—specifically, 3' sequencing versions 2 and 3, and Immune Profiling 5' sequencing version 1—leading to a wide range of expected fractions of multiplets. The expected multiplet fraction depends primarily on the number of cells loaded onto the encapsulation platform [4], and the heuristic gives a rate per 1,000 non-empty droplets depending on chemistry.

In Fig 3, heuristic estimations for multiplet fractions are illustrated as trend lines specific to each chemistry, with each dataset mapped onto its respective trend. It is important to note again, that cell hashing only detects multiplets formed between distinct samples, thus providing a lower bound for the actual multiplet fraction. The heuristic consistently underestimates the multiplet fractions, though the extent of this underestimation varies considerably. A notable example is the mouse kidney ('mkidney') dataset, with an observed multiplet fraction of about 37%, compared to a heuristic estimate of 17%, marking a striking 20% discrepancy. Conversely, the PBMC dataset ('pbmc') more closely aligns with its heuristic estimation, implying a potentially minimal impact from unidentified multiplets. However, as cell hashing remains uncommon, researchers cannot tell where their dataset lies on this spectrum.

To better estimate the true multiplet rate, we generalized the Poisson-based cell encapsulation model developed by Bloom [24] to accommodate multiple donors or samples per dataset. Unlike Bloom's original method, our model requires only total droplet counts and observed multiplet counts from cell hashing, assuming equal numbers of cells for each sample. This, while not as accurate as Bloom's model in cases with uneven sample contribution, makes the model more practical for publicly available datasets, where sample-specific information is rarely available. These estimates, shown as red points in Fig 3, aligned vertically with each dataset's heuristic and observed values, demonstrate, again, substantial variability. In the pbmc dataset, the difference is modest (about 2%) even though its significance is unknown, but in mkidney, the estimated rate approaches 60%, over 40 percentage points above the heuristic and 20 above the cell-hashing rate. These findings highlight a fundamental

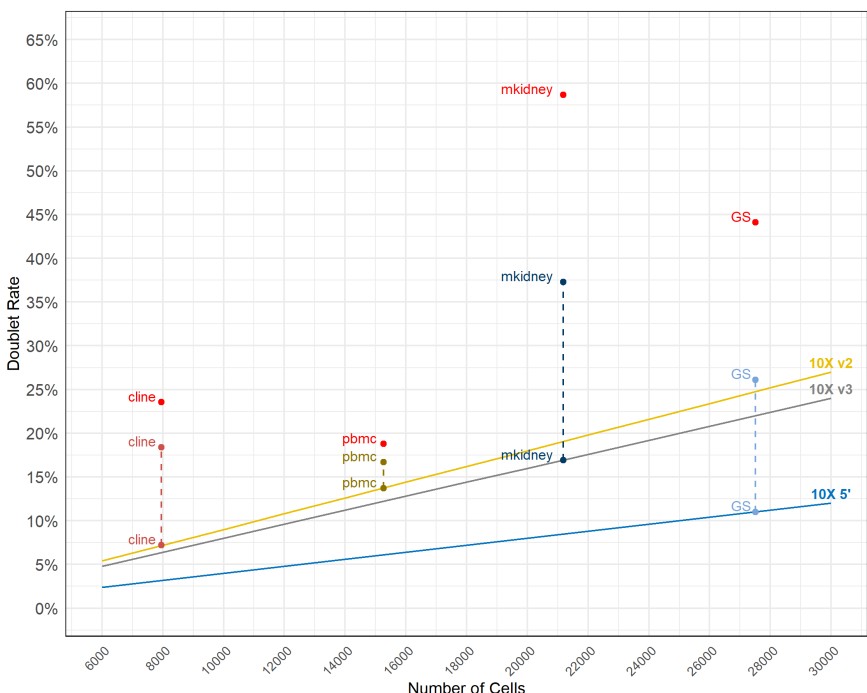

**Fig 3. Expected, observed and estimated true multiplet fractions.** Trend lines show the estimated fraction of multiplets per number of non-empty droplets observed in a 10X Genomics single-cell experiment, for the three chemistries used to generate the four datasets examined in the present manuscript. Each dataset is represented by three points: one on the trend line, showing the 10X Genomics estimation, another point connected to the previous point by a dashed line, showing the fraction of multiplets as indicated by the multiplets identified by cell hashing, and a third point in red, aligned vertically above the other two points for each dataset, showing the Poisson estimation for the true fraction of multiplets present in the dataset.

challenge in single-cell workflows: the proportion of droplets that should be excluded varies widely across datasets. In some cases, unidentified multiplets can be only a few hundreds of droplets; in others, they may comprise a substantial portion of the data, sometimes exceeding 50%. Most researchers lack a reliable method to establish the true number of multiplets in their datasets and might unknowingly retain significant numbers of multiplets in their analyses, potentially compromising analytical outcomes and reducing statistical power.

## Existing multiplet-detection strategies underperform

Given the large variation of multiplet fractions seen in the previous section, we assessed how many multiplets are identified by the eight commonly used multiplet-detection tools relative to the multiplet set defined by cell hashing. We compared the number, identity, and overlap of multiplets across methods for all four datasets. For clarity, we will focus in the main text and figures (starting with Fig 4) on the gold standard dataset as it is the largest and the most recent of the examined datasets. Corresponding analyses for the other datasets are reported in the supplemental figures S1 Fig–-S3 Fig.

Most remarkably, 73% of cell-hashing multiplets were not detected by any tool. These likely include homotypic multiplets, which are not detectable by any multiplet-detection tool. We stress that the tools are not at fault here because multiplet detection depends on the expected multiplet rate and we chose to use the heuristic-determined expected multiplet rates when

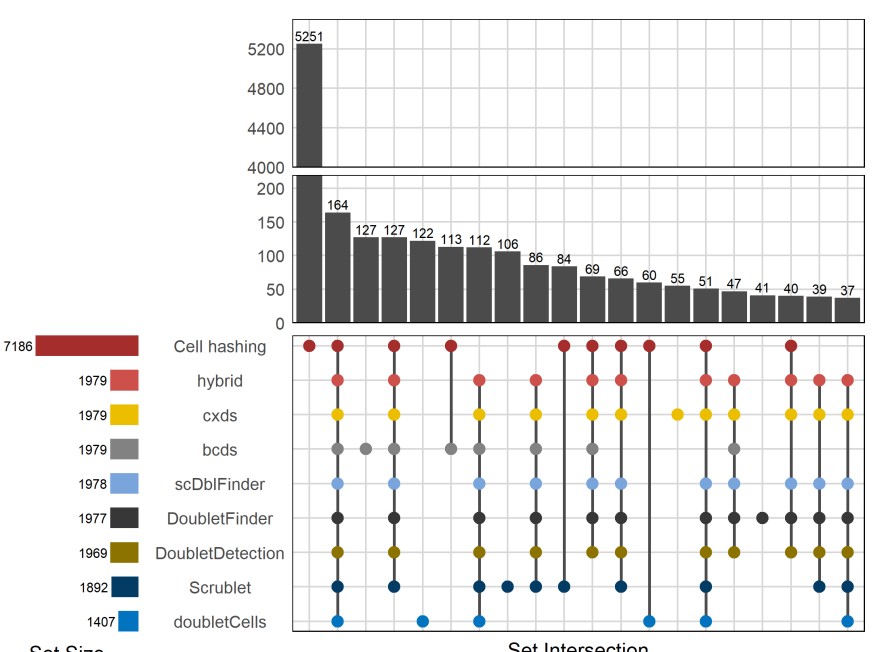

**Fig 4. Top 20 multiplet sets and intersections (gold standard dataset).** UpSet plot showing sets and set intersections of multiplets identified by the tools and by cell hashing in the gold standard dataset. Horizontal bars (bottom left) show the total number of multiplets identified by each method. Vertical bars (top right) show the sizes of the sets of detected multiplets. If a point is present for a method underneath a bar (bottom right plot panel), this means that this method has identified the particular multiplets that are members of the set denoted by that bar. If multiplet points are present underneath a bar, this means that multiple methods identified the same set of multiplets, i.e. their sets of multiplets intersect. Points are colored by method. Only the 20 largest multiplet sets and set intersections are shown. Note the y axis break.

running each tool—just as most users would. Using the observed or estimated true multiplet rates would most definitely improve the amount of detected multiplets, but such information is unavailable in non-multiplexed experiments.

Among the tools, bcds identifies the highest number of unique multiplets in this dataset. However, referring to S4 Fig, its accuracy is the lowest, implying that many of these multiplets are either false positives or intra-sample multiplets. Conversely, Scrublet, the most accurate tool by AUROC, finds fewer unique multiplets, but still misses most cell-hashing multiplets. In fact, only 164 out of 7,186 cell-hashing multiplets—just about 2.3%—were identified by all tools. This very limited intersection demonstrates substantial gaps in current computational detection.

We also investigated how similar cell-hashing multiplets are to those detected by multiplet-detection tools by visualizing their distribution in a two-dimensional projection of transcription space, using a UMAP plot (Fig 5). Cell-hashing multiplets are distributed in a UMAP region where most tools detect few, if any, multiplets. Multiplet-detection tools detect subset of multiplets that differ from those identified by cell hashing. The former identifies multiplets composed of different types of cells, regardless of whether they stem from the same sample or not, while cell hashing identifies multiplets that are formed between cells from different samples. These differences confirm that multiplet removal is not as simple as running a reputable tool and eliminating the identified multiplets. Different methods detect different subsets, and

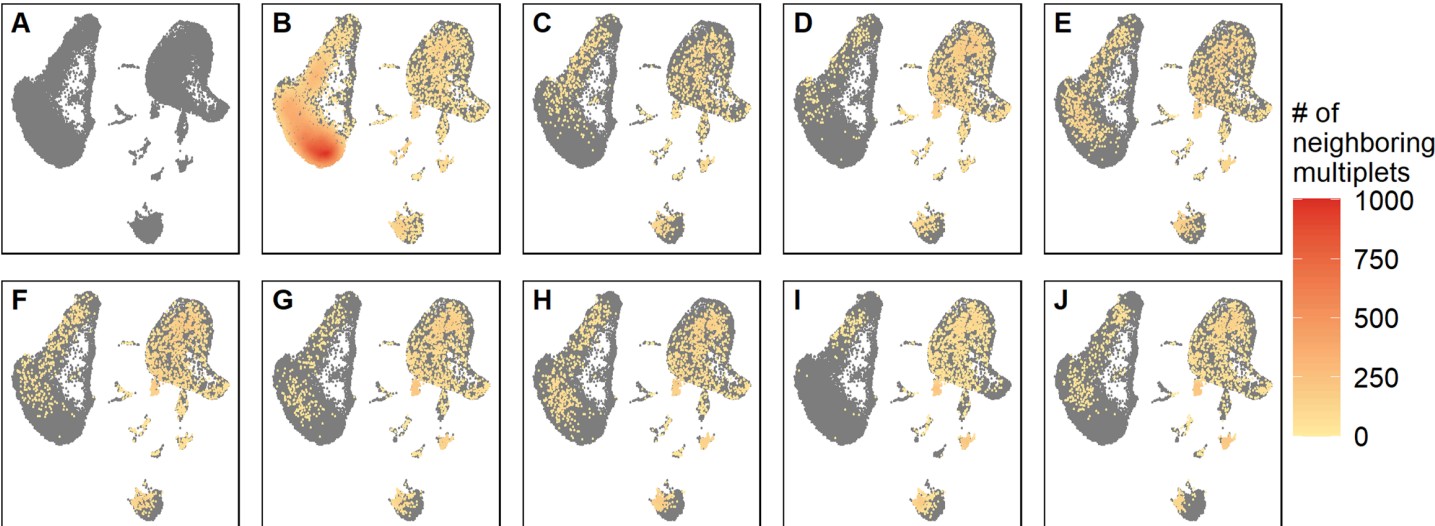

**Fig 5. Multiplet distribution and density across methods.** (A) UMAP of the gold standard dataset without removing any multiplets. UMAP of the gold standard dataset after removal of multiplets identified by (B) cell-hashing, (C) DoubletCells, (E) cxds, (E) bcds, (F) hybrid, (G) scDblFinder, (H) Scrublet, (I) DoubletDetection, and (J) DoubletFinder. Grey points are all droplets and colored points are multiplets, colored by the number of their multiplet neighbors within a fixed-sized neighborhood. Red to yellow indicate high to low density. The color scale is the same for all UMAPs.

their outputs vary substantially, with potentially significant consequences for downstream analysis.

Additionally, we observed that cell-hashing multiplets occupied the entire transcriptional space without forming exclusive clusters. This contradicts a common assumption that multiplets form separate clusters and suggests multiplets may significantly distort signals within otherwise homogeneous clusters, further complicating interpretations. That being said, there is always a possibility that one of the smallest clusters contains solely multiplets, but some were missed by cell hashing because they are from the same sample.

## Multiplets confound downstream analysis

We next assessed how unidentified multiplets affect clustering and cell-type annotation—two core components of single-cell analysis. Starting with clustering, the left panel of Fig 6 shows the average ranks across all four datasets and three different, routinely used clustering quality metrics (Calinski–Harabasz, Davies–Bouldin, and Silhouette value) [32–34]. We can see that "No processing" ranks worst overall, indicating that multiplet removal improves clustering quality, regardless of the assessed metric. Across all metrics and datasets, we found DoubletDetection, scDblFinder, and DoubletFinder to be the top performers, which aligns with the findings on synthetic datasets from the benchmarking by Xi and Li [7]. Cell hashing ranks fourth best, indicating that it also provides substantial improvements, albeit not as pronounced as the top three computational methods. This is not completely unexpected, since cell hashing only removes multiplets between cells from different samples, which are not necessarily the droplets that perturb clustering the most. As we have seen in Fig 4, cell hashing also removes a significantly higher number of multiplets than the computational tools, which could influence the metrics. However, this is not relevant here, since this analysis was done using stratified subsampling over 100 iterations per dataset—method combination before

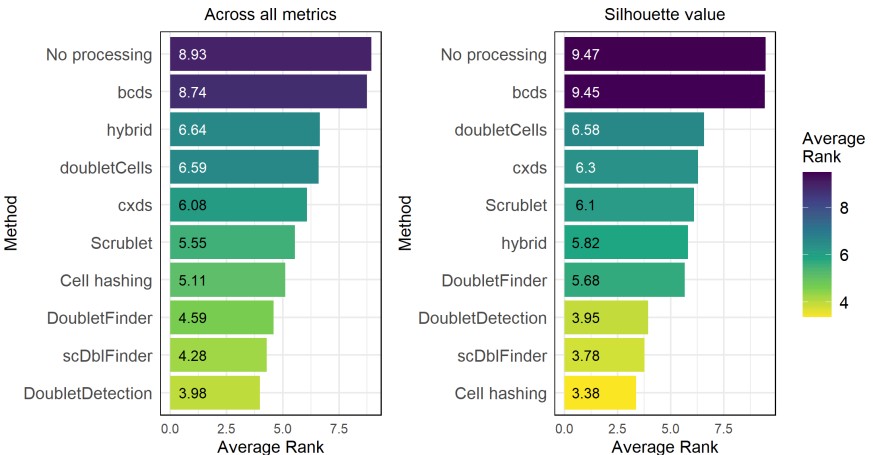

**Fig 6. Average ranking of multiplet removal methods by clustering quality.** Bar plots showing the average rank in clustering quality across all four datasets after multiplet removal by each of the eight tested methods and by cell hashing, compared to no processing. Left: Average ranks across all three clustering quality metrics assessed: Calinski–Harabasz Index, Davies–Bouldin Index, and Silhouette value. Right: Average ranks from the Silhouette value alone, one of the most widely used metrics for clustering quality in single-cell analysis. Ranks were computed after stratified subsampling of datasets (100 iterations, stratified by cluster) to a fixed number of droplets per dataset version, thereby controlling for differences in dataset size after multiplet removal. All the dataset, method, and metric combinations' values can be seen in supplemental figure S5 Fig. Lower average rank indicates better clustering quality.

computing the metrics, so that the number of droplets for each dataset version was kept constant. The right panel of Fig 6 shows the average ranks across all four datasets but this time only on the Silhouette value, which is commonly used to evaluate clustering quality and optimal cluster number in single-cell analysis. Interestingly, most results are similar to the left panel, with the exception of cell hashing, which now ranks best, DoubletDetection, which is now third-best, dropping two positions, and hybrid, which climbed three positions in comparison to its rank in the left panel. It is possible that removing cell-hashing multiplets has led to a representation of the underlying cell types that increases homogeneity within each cluster and heterogeneity between clusters, which the Silhouette value is more sensitive to. Looking at S5 Fig, however, which shows the values of all metrics for all dataset—method combinations, we can see that the results are quite variable and depending on the specific dataset and metric assessed, one can be led to different conclusions about the effectiveness of each method. The inconsistency between metrics highlights the challenge of evaluating clustering quality quantitatively in real datasets. We also have to note, that our clustering parameters were kept fixed across all runs and were not optimized in any way, meaning some tool-specific effects may have been missed.

We used scBubbletree [27] because it allows for a clear and quantitative visualization of cell clusters, their robustness and similarity, and their compositions (including multiplet fractions). In Fig 7, we compare bubble trees for the gold standard dataset with and without cell-hashing multiplets removed and with cell types annotated with scTab [28]. Both trees have high bootstrapping stability and a similar ladder-like structure, typical for PBMC datasets, where most cell types differentiate from a few common progenitors. However, two small (each close to 1% of the dataset) CD14 Monocyte clusters, 10 and 11, disappear after multiplet removal. This suggests that those clusters were artefactual, formed by multiplet-induced distortion rather than true biological signal. Additionally, we observe a noticeable redistribution of droplets between clusters, which is enough to change the relative sizes of the largest

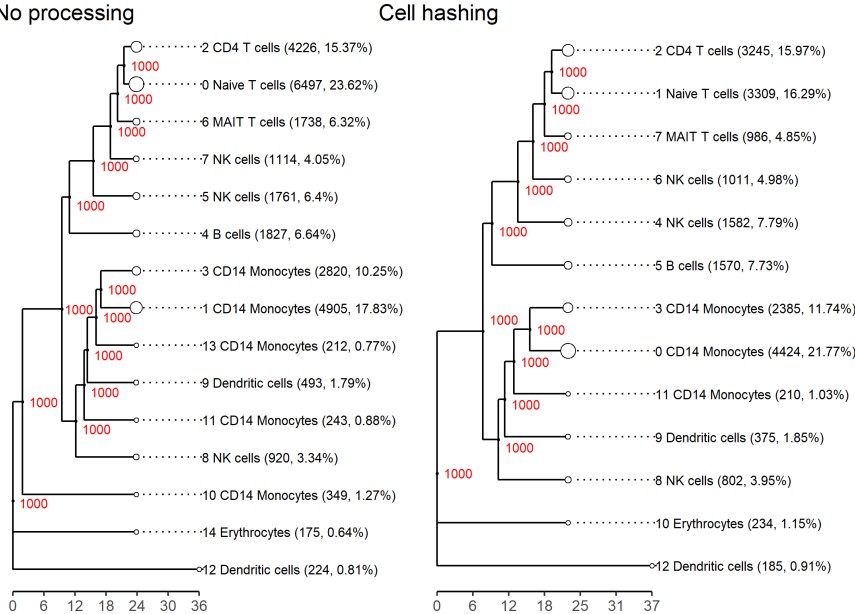

**Fig 7. Cell type annotation bubble tree comparison before and after multiplet removal.** Left: Gold standard dataset clustered with scBubbletree without removing multiplets and cluster cell types inferred with scTab. Right: Gold standard dataset clustered with scBubbletree after removing cell-hashing multiplets and cluster cell types inferred with scTab. Red numbers indicate the bootstrapping stability of the clusters, i.e. the number of times a cluster emerged out of 1000 bootstrap resamples of the data during clustering. The cell type annotations are shown next to each cluster, with the most numerous annotation in each cluster shown. Both dendrograms are extremely robustly clustered and only differ in the size of the clusters and the number of clusters. Note the ladder-like structure, indicating closely related clusters and the disappearance of the two CD14 Monocyte clusters in the left tree (clusters 10 and 11) that disappear in the right tree after multiplet removal, pointing to multiplet-induced distortion rather than true biology. Evidence for clusters 10 and 11 merging into other similar clusters after multiplet removal is also seen in the heatmap in Fig 10.

clusters. The previously second-largest cluster, cluster 1, becomes the largest after multiplet removal (6% size difference between the clusters before multiplet removal), indicating that multiplets can shift global cluster structure.

The bubble tree and stacked heatmaps in Fig 8 visualize multiplet fractions of each cell cluster. It shows the full bubble tree of the gold standard dataset without multiplet removal, along with heatmaps indicating the cluster distribution of multiplets identified by each multiplet removal method. As seen previously in Fig 5, multiplets are unevenly distributed across clusters, with notable disagreement between the tools and cell hashing. Cell hashing places approximately 43% of its identified multiplets in cluster zero, whereas all tools identify very few multiplets there, with the highest percentages being around 5% (bcds and Scrublet). This mismatch likely stems from the tools' focus on heterotypic multiplets, as many of the cell-hashing multiplets in cluster zero may be homotypic and thus hard to detect computationally. Regardless, a substantial number of undetected multiplets in cluster zero are misclassified as singlets, potentially distorting interpretability. The large concentration of cell-hashing multiplets in cluster zero also explains why this cluster shrinks after multiplet removal, as observed in Fig 7. Meanwhile, most tools assign the majority of their multiplets to clusters one and three, which are only moderate in size by cell-hashing standards. Overall, cell hashing places the overwhelming majority of multiplets in the upper part of the tree, while tools detect them

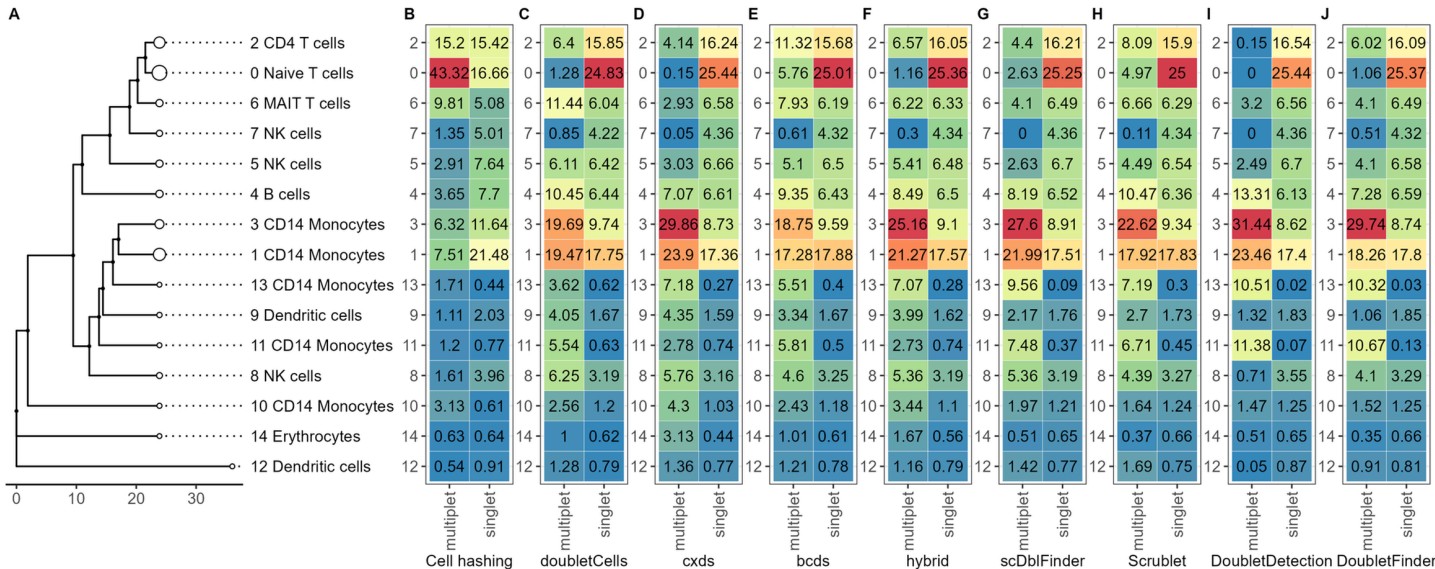

**Fig 8. Dataset-level distribution of singlets and multiplets across clusters by method.** (A) Annotated bubble tree of the gold standard dataset. Heatmap of the gold standard dataset indicating putative singlets and multiplets identified by (B) cell hashing, (C) DoubletCells, (D) cxds, (E) bcds, (F) hybrid, (G) scDblFinder, (H) Scrublet, (I) DoubletDetection, and (J) DoubletFinder. Each column integrates to 100, showing the distribution of multiplets or putative singlets across clusters as a percentage. Red to blue indicate high to low percentage. The color scale is the same for all heatmaps.

more in the middle, within a distinct major branch, revealing a clear disagreement between the distributions of actual multiplets and droplets classified as multiplets.

Using scBubbletree's horizontal integration, we generated Fig 9, a flipped-axis version of Fig 8, to examine the internal composition of each cluster, i.e. the fraction of each cluster classified as a multiplet or a singlet. The disagreement between tools and cell hashing remains clear, but new insights emerge. For instance, cluster zero appears to comprise roughly a 50–50 mix of multiplets and singlets by cell hashing, while most tools assign over 90% of this cluster's droplets as singlets. Clusters one and three now show more similar internal compositions across methods, softening the previously observed discrepancies. Additional differences arise in clusters two, six, and ten, where cell hashing detects much higher internal multiplet fractions than the tools. These undetected multiplets could distort biological interpretation depending on their transcriptional profiles. The reverse issue of false positives is also present. Small clusters like 11 and 13, according to DoubletFinder and DoubletDetection, are almost entirely composed of multiplets, despite low cell-hashing multiplet counts. While there is uncertainty whether these small clusters could actually be comprised of a high fraction of intra-sample multiplets, which are impossible to detect with cell hashing, there remains a substantial risk that they represent false positives. In that case, overcalling risks removing rare or biologically relevant cell types or state, underscoring how both false negatives and false positives in multiplet detection can mislead downstream analysis.

The stability of cell-type annotations is also another interesting point to consider. Returning back to Fig 7, it is evident that annotations remained largely stable, which holds true even when multiplets were removed based on other tools (S6 Fig). The most notable observation is that five CD14 Monocyte clusters merge into three after removing cell-hashing multiplets. This shift results from droplet redistribution rather than annotation changes and suggests the elimination of artefactual clusters. Such reorganization was not observed when removing multiplets identified by tools (S6 Fig), with the exception of cxds and hybrid (which blends

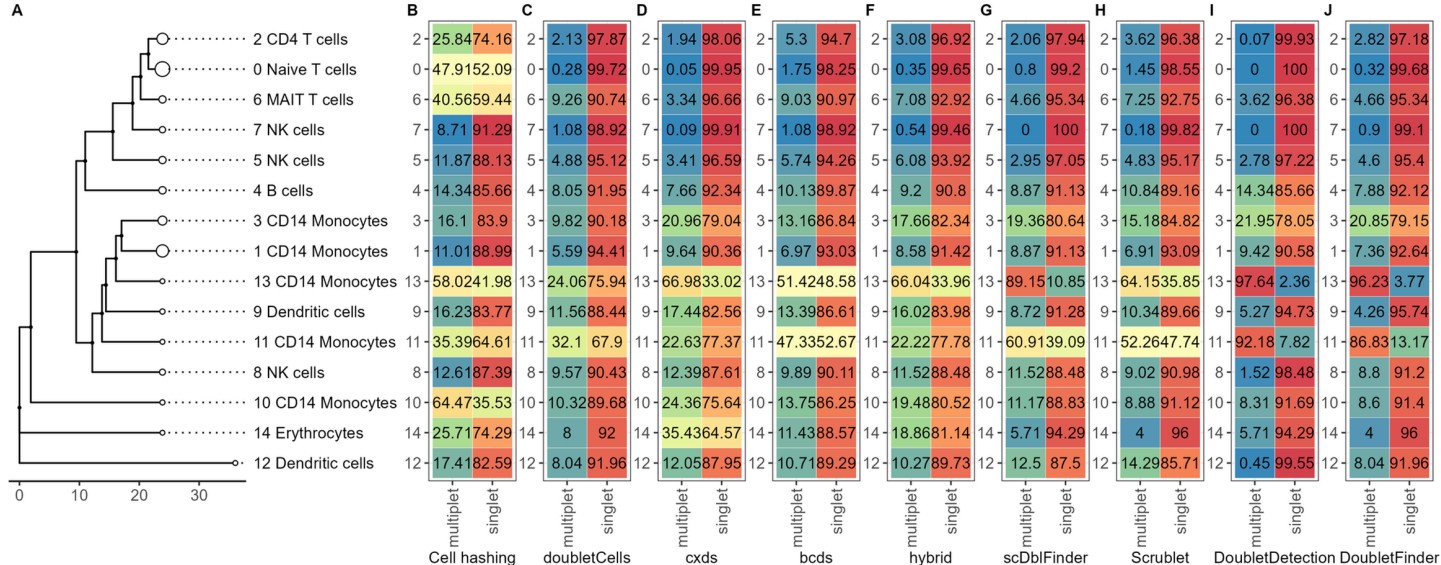

**Fig 9. Cluster-level composition of singlets and multiplets by method.** (A) Annotated bubble tree of the gold standard dataset. Heatmap of the gold standard dataset indicating putative singlets and multiplets identified by (B) cell hashing, (C) DoubletCells, (D) cxds, (E) bcds, (F) hybrid, (G) scDblFinder, (H) Scrublet, (I) DoubletDetection, and (J) DoubletFinder. Each row per method integrates to 100, showing the percentage of multiplets and putative singlets within each cluster as a percentage. Red color to blue indicate high to low percentage. The color scale is the same for all heatmaps.

cxds and bcds). Although scTab cluster annotations are robust, the disappearance of clusters after multiplet removal highlights their confounding role: they may fragment coherent cell populations, inflate diversity, or produce artefactual clusters. Even if robust annotation tools are employed, these changes can affect interpretation and lead to false discovery of nonexistent cell types or cell states.

Because of the aforementioned observations, we decided to outline how putative singlets from the unprocessed dataset were reassigned in the bubble tree after removing cell-hashing multiplets. This will allow us to confirm stable cluster membership but, more importantly, to understand how droplets from disappeared clusters are redistributed and whether we have droplet reshuffling or cluster consolidation. Fig 10 shows how many droplets pre-removal (horizontal axis) and post-removal (vertical axis) clusters have in common, i.e. the number of droplets that remained in the same cluster and the number of droplets that were reassigned cluster after multiplet removal. It is apparent, that most droplets remain in the same cluster, with over 95% retention in most cases. The exceptions are four small pre-removal clusters, 10, 11, and 13, which are CD14 Monocytes and cluster 14 which is comprised of Erythrocytes (here meaning erythroid progenitors, since mature erythrocytes have no nucleus or RNA). Clusters 13 and 10, both CD14 Monocyte clusters, show the most significant redistribution as is apparent by the low Jaccard similarity between the pre-removal and post-removal clusters. Cluster 13 primarily merges with cluster 11—another CD14 Monocytes cluster—indicating consolidation of biologically related groups under a single major cell type. The remaining handful of droplets from cluster 13 are reassigned to clusters three and zero, also CD14 Monocyte clusters. Pre-removal cluster 10 largely maps to post-removal cluster 10, itself mostly derived from pre-removal cluster 14 with the same annotation. Some minor droplet scattering occurs across lymphocyte clusters, but largely between transcriptionally similar populations. Post-removal cluster 11 predominantly consists of droplets from

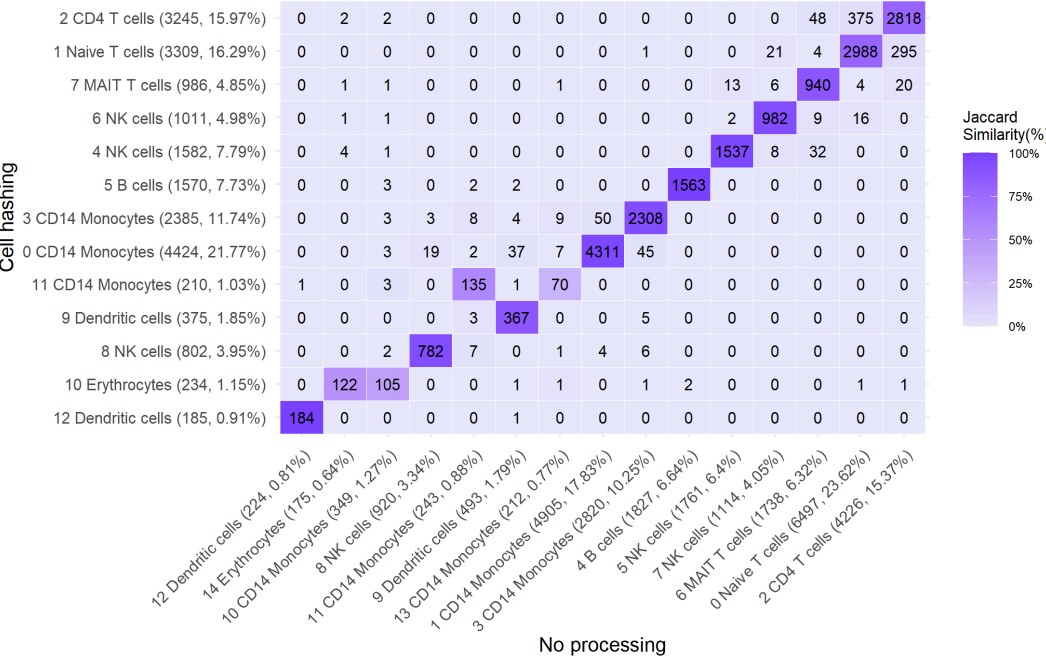

**Fig 10. Cluster correspondence before and after multiplet removal.** Heatmap of droplets shared between clusters before and after multiplet removal. On the horizontal axis the clusters before removing any multiplets are shown in the order they appear on the respective bubble tree. On the vertical axis the clusters after removing any multiplets are shown in the order they appear on the respective bubble tree from right to left. The numbers show the droplets that are common between each pair of clusters. Heatmap tiles are coloured based on the Jaccard similarity between each pair of clusters. Cell-hashing multiplets were excluded for this figure. The overwhelming majority of clusters remain stable after multiplet removal, with only a few clusters merging and some minor droplet reshuffling between clusters. Note how pre-multiplet-removal clusters 14 and 10 consolidate to post-multiplet-removal cluster 10, while pre-multiplet-removal clusters 11 and 13 consolidate to post-multiplet-removal cluster 11. In both cases the cell types are related and the consolidation is likely due to multiplet-induced transcriptional noise.

pre-removal cluster 11 and to a lesser extend from cluster 13 in a roughly 2:1 ratio, while post-removal cluster 10 mainly consists of droplets from clusters 10 and 14 in roughly equal proportions. Most other clusters retain their original composition after cell-hashing multiplet removal, with any droplet redistribution limited to closely related clusters. The largest droplet exchange in the dataset is an example of redistribution between closely related clusters and occurs between pre-removal clusters zero (naive T cells) and two (CD4 T cells). Post-removal, the majority of their droplets are incorporated into clusters one and two, respectively, and the exchange between them involves a few hundred droplets. Since annotation is assigned per droplet, not per cluster, the similarity between clusters two and zero reflects a mixture of T cell types within both. Their annotations simply reflect the most numerous label within each cluster, which explains the minor label reassignments seen after cell-hashing multiplet removal. Cluster mixing, consolidation, and droplet reshuffling could indicate the presence of unidentified multiplets, which undermine the interpretability and reliability of otherwise robust clustering and annotation tools. By extension, the cases illustrated above show that multiplet removal can improve the clarity and reliability of clustering and annotation, even when the overall impact appears modest. We emphasize that cell-hashing multiplets only constitute a lower bound of the true multiplet count, and as demonstrated, not just how many but which multiplets are removed can substantially affect downstream analysis.

To assess how multiplets influence another important downstream task, namely differential gene expression (DGE), we compared two DGE analysis runs, highlighting the differences in differentially expressed genes (DEGs) between pre-multiplet-removal clusters "1 CD14 Monocytes" and "13 CD14 Monocytes", before and after cell-hashing multiplet removal, without reclustering. In Fig 11, between the left and middle volcano plots, we see that the majority of DEGs are preserved after multiplet removal, and mostly the effect sizes impacted. The top ten most strongly differentially expressed genes are annotated in red, and canonical markers for CD14 monocytes are highlighted in black. Notably, NK/cytotoxic markers (NKG7, PRF1, CST7, GZMA, GNLY, CCL5) are among the strongest signals when multiplets are included but lose significance and show attenuated effect sizes after multiplet removal, while still remaining the top ranking genes by $\log_2$ fold change. CD 14 monocyte canonical markers seem to not be affected by multiplet removal, which we take as a sign of a true biological signal, albeit with apparent residual multiplet contamination. Assuming cell-type annotation is to be trusted, this pattern is consistent with heterotypic monocyte—NK cell multiplets being unevenly distributed across the two CD14 monocyte clusters examined. Removing cell-hashing multiplets reduces—but does not completely remove—this effect, leaving residual NK cell signal, which likely reflects undetected intra-sample multiplets. This illustrates how DGE is particularly vulnerable to multiplet contamination, even when both clustering and cell-type annotation seem well resolved. Examining the rightmost panel of Fig 11, we see that despite $\log_2$ fold changes between the two DGE runs being highly correlated (Spearman = 0.96), the set of significant DEGs only partially overlapped (Jaccard = 0.72). We can thus conclude, that in comparisons of closely related clusters, multiplets can both elevate false-positive

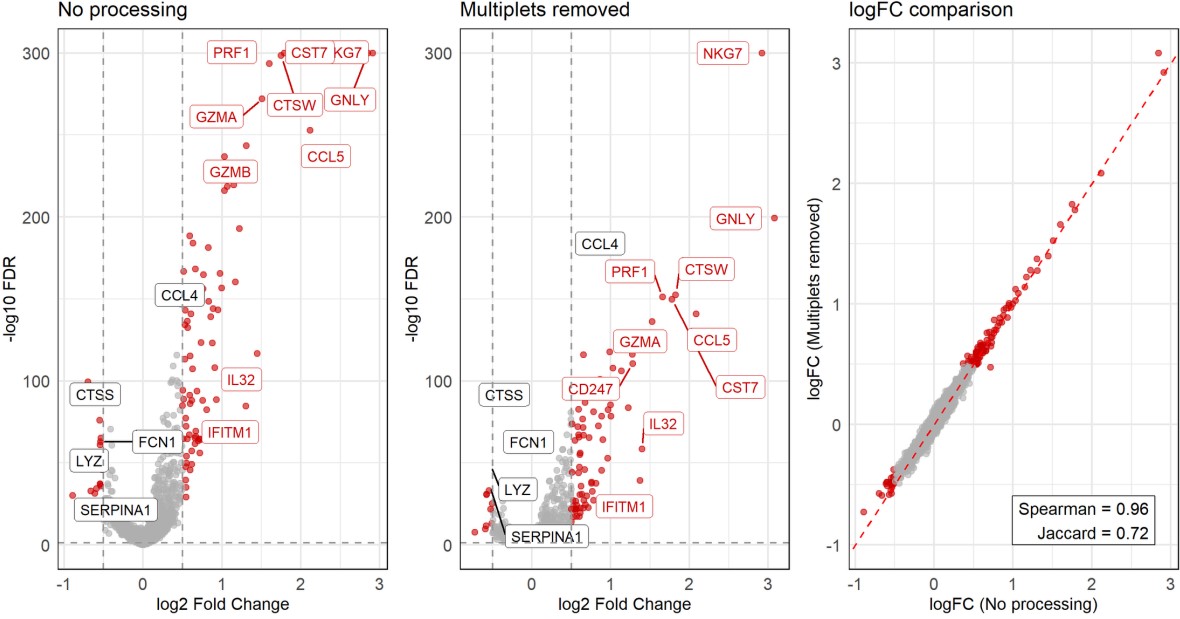

**Fig 11. Differential gene expression analysis.** Differential gene expression (DGE) between pre-multiplet-removal clusters "1 CD14 Monocytes" and "13 CD14 Monocytes" was evaluated using MAST before (left, "No processing") and after removing cell-hashing multiplets (middle, "Multiplets removed", no further processing). Volcano plots show $\log_2$ fold change (x-axis) versus $-\log_{10}$ FDR (y-axis), with significant genes highlighted in red. Canonical markers for CD14 monocytes that were identified as significant in the analysis are also shown, in black. While most DEGs are preserved across conditions, several genes lose or gain significance after multiplet removal, and effect sizes shift for a subset of markers. The rightmost panel compares $\log_2$ fold changes between the two DGE runs (with and without multiplets), with high overall concordance (Spearman = 0.96) but only partial overlap of significant genes (Jaccard = 0.72).

DEGs and exaggerate effect sizes, whereas the underlying monocyte contrast remains largely preserved.

## Discussion

This manuscript highlights the pervasive and underestimated impact of multiplets in single-cell RNA sequencing (scRNA-seq) datasets even after application of dedicated tools for multiplet removal. Our findings confirm that undetected multiplets are not only much more frequent than previously recognized but also that they can confound clustering and cell-type annotation. We show that the commonly used heuristic for estimating the number of multiplets to expect in a given experiment, that is based on the number of droplets retrieved, systematically underestimates the amount of multiplets present in the dataset. While the gold standard dataset with about 26% cell-hashing multiplets might not be representative of older single-cell studies, it reflects the current high-throughput practices where such multiplet rates are increasingly common with more effort put into raising throughput and yield, and it also falls well within the multiplet range across our other datasets (16–37%). The gold standard dataset was chosen not because it is representative of the average scRNA-seq experiment but because it uniquely combines multimodal data with cell-hashing, which holds new information about multiplets. We were also able to demonstrate that commonly used multiplet-detection tools underperform in identifying the full extent of multiplets, with discrepancies varying across datasets, sometimes exceeding multiplet rates almost three times that of the heuristic prediction. The Poisson-based estimation we provide in this work can help illustrate how multiplet rates can in reality be significantly higher than previously thought, underscoring the risk of relying on conventional practices for multiplet identification and removal.

Even the best-performing tools, such as Scrublet and DoubletFinder, identify only a small fraction of the known multiplets in the tested datasets, and no tool has been consistently superior across all datasets. The minimal overlap between cell-hashing multiplets and tool-identified multiplets adds to the evidence that these methods detect fundamentally different subsets of multiplets. The inability of any method to detect the entire set of multiplets in any given dataset is a critical limitation, as multiplets can still interfere with gene expression variability, clustering, and cell-type annotation. Given that many scRNA-seq studies analyze closely related cell types, the failure to remove homotypic multiplets could be particularly problematic in studies with a dominating cell type, like cell atlases and developmental/differential biology.

Clustering results indicate that multiplets are interspersed throughout all clusters rather than forming multiplet-only clusters, complicating their removal. While some pre-removal clusters contained a majority of multiplets, these were not lost after removal but redistributed and merged with neighboring clusters, with improved Silhouette values suggesting increased cluster cohesion. This indicates that multiplets, at least in some cases, do not form purely spurious, multiplet-only clusters but instead inflate heterogeneity within existing cluster structure. Removing cell-hashing multiplets led to refined cluster boundaries, consistent with a modest improvement in clustering quality rather than wholesale elimination of cell types. The redistribution of droplets between some clusters after multiplet removal suggests that certain cell types may be disproportionately affected by multiplets. We particularly highlight the risk of arriving to false biological conclusions due to the presence of artificial cell-type heterogeneity in datasets of closely related cell types. Conversely, overly aggressive multiplet calling risks the removal of rare or biologically relevant cell states. This was observed in the gold standard dataset, in small clusters where some of the multiplet-removal tools labeled nearly all droplets

as multiplets. This highlights how false-positive multiplet calls, just as much as false negatives, can lead to potential pitfalls in downstream interpretation. Beyond clustering, multiplets can also distort differential gene expression analysis. In a proof-of-concept comparison we observed that while the majority of DEGs were preserved after multiplet removal, effect sizes and significance levels shifted for a subset of genes, consistent with multiplets introducing misleading variability. This further reinforces the risk of misinterpreting expression changes between closely related cell states.

Our findings suggest that the presence of multiplets in single-cell data should get greater attention from active research. A combination of approaches, such as cell hashing, multimodal multiplet identification tools, and custom filtering techniques should be considered for multiplet identification and removal. Given their rapid progress it is conceivable that AI models may in the future contribute to a more thorough multiplet removal. Until then, we advise caution and skepticism when interpreting scRNAseq data and drawing conclusions about putative novel cell types or cell states, as these may be artifacts of multiplet contamination.

## Limitations

We acknowledge several limitations in the present work. Firstly, our estimation of the true multiplet rate relies on the classical Poisson droplet-loading model, which assumes cells are independently and identically captured and thus droplets can form multiplets with equal probability [4,13,35,36]. However, peer-reviewed literature documents cell-type–specific co-encapsulatation biases driven by cell size, viability, adhesion, and stable in-vivo interactions, indicating that this assumption can be violated [37–40]. We did not directly compare doublets (two cells in a single droplet) versus higher-order multiplets (three or more cells in a single droplet), and while Poisson statistics predict higher-order multiplets should be rare, their prevalence could be elevated under the intentionally high cell-loading that is typical of cell-hashing experiments. Higher-order multiplets are indeed more transcriptionally heterogeneous and potentially more disruptive, which in principle makes them easier to detect by multiplet-removal tools. However, most computational tools are optimized for doublet detection, whereas cell hashing is agnostic to multiplet order. We limited our analysis to 10X Genomics' technologies (3' v2, 3' v3, and 5' Immune Profiling v1), since they are the most widely used single-cell methods at present. Therefore our datasets are exclusively droplet-based, and our results may not generalize to other single-cell technologies. We also, due to a lack of alternatives, rely on cell hashing as a quasi-ground truth to compare against, which is only a lower bound for the number of multiplets present in the dataset. Another limitation is that we wanted to emulate multiplet removal workflows according to the prevailing practice and therefore concentrated on the most widely used tools (which are still among those tested in the benchmarking study [7]). We therefore did not consider newer and potentially better-performing but less frequently used tools. Finally, a major limitation of this manuscript is that we completely ignored the layers of information other than transcriptional information, which are provided by the gold standard dataset and which made us nickname it 'gold standard' in the first place. We decided that the further investigation of multimodal approaches to infer multiplets is outside the scope of the present study and belongs to a separate work that will attempt to develop a tool to take advantage of the extra information the gold standard provides.

## Supporting information

**S1 Table. Overview of dataset information.**
(PDF)

**S1 Fig. Top 20 multiplet sets and intersections (cline dataset).** UpSet plot showing sets and set intersections of multiplets identified by the tools and by cell hashing in the cline dataset. Horizontal bars (bottom left) show the total number of multiplets identified by each method. Vertical bars (top right) show the sizes of the sets of detected multiplets. If a point is present for a method underneath a bar (bottom right plot panel), this means that this method has identified the particular multiplets that are members of the set denoted by that bar. If multiplet points are present underneath a bar, this means that multiple methods identified the same set of multiplets, i.e. their sets of multiplets intersect. Points are colored by method. Only the 20 largest multiplet sets and set intersections are shown. Note the y axis break.
(TIF)

**S2 Fig. Top 20 multiplet sets and intersections (mkidney dataset).** UpSet plot showing sets and set intersections of multiplets identified by the tools and by cell hashing in the mkidney dataset. Horizontal bars (bottom left) show the total number of multiplets identified by each method. Vertical bars (top right) show the sizes of the sets of detected multiplets. If a point is present for a method underneath a bar (bottom right plot panel), this means that this method has identified the particular multiplets that are members of the set denoted by that bar. If multiplet points are present underneath a bar, this means that multiple methods identified the same set of multiplets, i.e. their sets of multiplets intersect. Points are colored by method. Only the 20 largest multiplet sets and set intersections are shown. Note the y axis break.
(TIF)

**S3 Fig. Top 20 multiplet sets and intersections (pbmc dataset).** UpSet plot showing sets and set intersections of multiplets identified by the tools and by cell hashing in the pbmc dataset. Horizontal bars (bottom left) show the total number of multiplets identified by each method. Vertical bars (top right) show the sizes of the sets of detected multiplets. If a point is present for a method underneath a bar (bottom right plot panel), this means that this method has identified the particular multiplets that are members of the set denoted by that bar. If multiplet points are present underneath a bar, this means that multiple methods identified the same set of multiplets, i.e. their sets of multiplets intersect. Points are colored by method. Only the 20 largest multiplet sets and set intersections are shown. Note the y axis break.
(TIF)

**S4 Fig. AUROC-based accuracy of multiplet detection across datasets and methods.**
(TIF)

**S5 Fig. Effect of multiplet removal on clustering quality across datasets and methods.** Bar plot of clustering quality after the removal of multiplets detected by each of the eight methods tested and by cell hashing. Clustering was evaluated using three complementary metrics: the Calinski–Harabasz Index (higher is better; rewards global separation and a higher number of compact clusters), the Davies–Bouldin Index (lower is better; rewards compact clusters that are well-separated from their nearest neighbors), and the Silhouette value (closer to 1 is better; rewards droplets being closer to their own cluster than to other clusters). To account for differences in dataset size, we fixed the number of droplets per dataset to a common value—

rounded down to just below the smallest dataset—and performed stratified subsampling, repeating this procedure 100 times per method and metric. Bars show the mean clustering quality, and error bars denote variability across subsamples.
(TIF)

**S6 Fig. Bubble tree comparison before and after multiplet removal across all methods.**
(A) Gold standard dataset clustered with scBubbletree without removing multiplets. (B) Gold standard dataset clustered with scBubbletree after removing cell-hashing multiplets. (C) Gold standard dataset clustered with scBubbletree after removing DoubletCells multiplets. (D) Gold standard dataset clustered with scBubbletree after removing cxds multiplets. (E) Gold standard dataset clustered with scBubbletree after removing bcds multiplets. (F) Gold standard dataset clustered with scBubbletree after removing hybrid multiplets. (G) Gold standard dataset clustered with scBubbletree after removing scDblFinder multiplets. (H) Gold standard dataset clustered with scBubbletree after removing Scrublet multiplets. (I) Gold standard dataset clustered with scBubbletree after removing DoubletDetection multiplets. (J) Gold standard dataset clustered with scBubbletree after removing DoubletFinder multiplets. The robustness of clustering was assessed with bootstrapping, i.e. by the number of times a cluster emerged out of 1000 bootstrap resamples of the data during clustering. Red asterisks denote the only branches with a bootstrapping value less than 1000.
(TIF)

## Author contributions

**Conceptualization:** Dimitris Ttoouli.

**Data curation:** Dimitris Ttoouli.

**Formal analysis:** Dimitris Ttoouli.

**Funding acquisition:** Daniel Hoffmann.

**Investigation:** Dimitris Ttoouli.

**Methodology:** Dimitris Ttoouli.

**Project administration:** Daniel Hoffmann.

**Resources:** Daniel Hoffmann.

**Supervision:** Daniel Hoffmann.

**Validation:** Dimitris Ttoouli.

**Visualization:** Dimitris Ttoouli.

**Writing – original draft:** Dimitris Ttoouli.

**Writing – review & editing:** Dimitris Ttoouli, Daniel Hoffmann.

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
