## [Decision Letter · Decision Letter 0]

9 Jul 2025

PONE-D-25-32117Multiplets in scRNA-seq data: extent of the problem and efficacy of methods for removalPLOS ONE

Dear Dr. Ttoouli,

Thank you for submitting your manuscript to PLOS ONE. After careful consideration, we feel that it has merit but does not fully meet PLOS ONE’s publication criteria as it currently stands. Therefore, we invite you to submit a revised version of the manuscript that addresses the points raised during the review process.

We look forward to receiving your revised manuscript.

Kind regards,

Nagarajan Raju

Academic Editor

PLOS ONE

Journal Requirements: 

 [This study was funded in part by grant HO 1582/12 from Deutsche Forschungsgemeinschaft.]. 

Additional Editor Comments:

I suggest authors to go through the comments from the reviewer and address them in the revised version of the manuscript

Reviewers' comments:

Reviewer's Responses to Questions

**Comments to the Author**

1. Is the manuscript technically sound, and do the data support the conclusions?

Reviewer #1: Yes

2. Has the statistical analysis been performed appropriately and rigorously? 

Reviewer #1: Yes

3. Have the authors made all data underlying the findings in their manuscript fully available?

Reviewer #1: Yes

4. Is the manuscript presented in an intelligible fashion and written in standard English?

Reviewer #1: Yes

5. Review Comments to the Author

Reviewer #1: Single-cell RNA sequencing (scRNA-seq) has revolutionized the field of transcriptomics, allowing for in-depth analysis of individual cells. However, multiplets—droplets containing more than one cell—are a known artifact that can significantly impact the accuracy and interpretation of scRNA-seq data. This study comprehensively evaluates the prevalence of multiplets across diverse datasets and assesses the effectiveness of commonly used detection tools. The authors utilize cell hashing as a benchmark to determine the true multiplet rate and refine a Poisson-based model to estimate multiplet frequencies.

Major:

1, Is the assessment of multiplet removal limited to specific sequencing technologies? What is the typical prevalence of multiplets across datasets? Please provide supplemental information about the datasets used (e.g., cell types, sequencing platforms, library preparation methods).

2, Can the "gold standard" dataset with a 26.13% multiplet rate be considered representative of typical scRNA-seq studies? Furthermore, do multiplets (capturing >2 cells) exhibit more pronounced distinguishing features compared to doublets (2 cells), and if so, how does this impact detection?

3, The observation that the unprocessed data ("No processing") does not consistently yield the worst clustering performance (Fig. 6) requires further investigation. Additionally, could the improved performance after multiplet removal be partially confounded by the reduced cell count? Please address how changes in dataset size post-removal might influence clustering metric comparisons.

4, The analysis of clustering results should be extended:

a) Does multiplet removal eliminate spurious clusters predominantly composed of multiplets?

b) Conversely, could it potentially hinder the discovery of rare, biologically relevant cell types or states?

c) Beyond clustering, what impact does multiplet contamination (and its removal) have on other critical downstream analyses (e.g., differential expression, trajectory inference, cell-cell communication)?

6. PLOS authors have the option to publish the peer review history of their article (what does this mean?). If published, this will include your full peer review and any attached files.

Reviewer #1: No

---

## [Author Response · Author response to Decision Letter 1]

25 Aug 2025

Dear Editor, dear Reviewer,

We thank you both for the constructive comments on our manuscript. We have carefully revised the text and figures in response. Below we provide a point-by-point response to Reviewer #1’s comments, and also a mark-up version of the manuscript with changes to the first version of the manuscript highlighted. Line numbers given in our responses below refer to the consolidated revised manuscript, not the mark-up version.

Reviewer #1 – Major Comments

1. “Is the assessment of multiplet removal limited to specific sequencing technologies? What is the typical prevalence of multiplets across datasets? Please provide supplemental information about the datasets used (e.g., cell types, sequencing platforms, library preparation methods).”

We have altered the text in the Introduction (page: 2, lines: 12-14, 18-21) to clarify that multiplets are a universal phenomenon in single-cell technologies, but particularly prominent in droplet-based methods, and we report typical prevalences of multiplets in general. In the Datasets section of Methods (page: 3, lines: 83-85), we have changed phrasing so as to be explicit about the sequencing technology and platform. The dataset-specific information was already provided in the same section (page: 3-4, lines: 86-116), but for clarity we have also created a new Supplemental Table S1 summarizing source, tissue and number of samples, library preparation chemistry, multiplexing method, droplet counts, observed multiplet counts, and resulting multiplet rate for all datasets analyzed for this work. We now explicitly state in the Limitations section of the Discussion (page: 16, lines: 608-611), that our analysis is restricted to droplet-based scRNA-seq using 10x Genomics chemistries (3′ v2, 3′ v3, and 5′ Immune Profiling v1) and may not generalize to other platforms.

2. “Can the ‘gold standard’ dataset with a 26.13% multiplet rate be considered representative of typical scRNA-seq studies? Furthermore, do multiplets (>2 cells) exhibit more pronounced distinguishing features compared to doublets, and if so, how does this impact detection?”

The gold-standard dataset was chosen not because it is representative of the average scRNA-seq experiment but because it uniquely combines multimodal data with cell-hashing, which holds new information about multiplets. We added a clarification in the Discussion (page: 15, lines: 539-546) noting that ~26% is not representative of older studies but reflects current high-throughput droplet practice and that its rate falls within our observed range across all datasets (16–37%; Fig. 3). Regarding the comparison between doublets and multiplets, we have clarified this point in the Limitations section of the Discussion (page: 16, lines: 600-608). Specifically, we acknowledge that while Poisson statistics predict higher-order multiplets should be rare, their prevalence may be somewhat elevated under the intentionally high cell-loading conditions typical of cell-hashing experiments. Such higher-order multiplets are more transcriptionally heterogeneous and potentially more disruptive, which in principle makes them easier to detect. However, most computational tools are optimized for doublet detection, whereas cell hashing is agnostic to multiplet order.

3. “The observation that unprocessed data does not consistently yield the worst clustering performance requires further investigation. Could the improved performance after multiplet removal be partially confounded by the reduced cell count?”

We thank the reviewer for bringing this important point to our attention. We agree that dataset size differences could indeed confound clustering quality comparisons. To address this, we performed stratified subsampling prior to metric calculation, fixing all datasets to a common cell count (just below the smallest dataset size). We repeated subsampling 100 times per dataset and method combination, and calculated clustering metrics on the resulting subsets. Error bars (IQR) were very small and one could not judge whether they are large enough to change the ranking of best-to-worst performing method, so we decided to put the revised version of Figure 6 with error bars in Supporting information as Figure S5 (previous Figure S5 is now Figure S6 due to order of appearance), for completeness. We then summarized performance by computing average ranks across all three metrics, as well as separately for the Silhouette index, which is commonly used to evaluate clustering quality and optimal cluster number in scRNa-seq. This analysis confirmed that the “No processing” condition consistently performs worst, across datasets and across clustering metrics. Besides the new figure, the new findings are reflected in the manuscript with a new "Clustering quality assessment" section in Methods (page: 7, lines: 214-234), and a new Results section detailing these findings in the new figure 6 (pages: 10-11, lines: 369-400).

4. “The analysis of clustering results should be extended:

a) Does multiplet removal eliminate spurious clusters predominantly composed of multiplets?"

We have addressed this point directly in the manuscript. In the gold-standard dataset, only two pre-removal clusters (10 and 13) contained a majority of multiplets (64.5% and 58% cell-hashing multiplets, respectively). After removal, however, these clusters were not eliminated but redistributed and merged with adjacent, closely related clusters. In Fig. 10 we point out which cluster pairs fused into new, single clusters after multiplet removal and in Fig. 9 we show the bubble tree that demonstrates how these clusters are closely related to each other and that, with the exception of the two aforementioned clusters, no cluster comprises more than 50% cell-hashing multiplets. Silhouette values improved after multiplet removal, indicating increased cluster cohesion rather than wholesale elimination of cell types. We have changed the wording of this conclusion for clarification in the Discussion (page: 15-16, lines: 564-571). Uncertainty remains for very small clusters that could consist of undetected multiplets, which we note in the Results as a caveat (page: 12, lines: 445-448).

b) Conversely, could it hinder the discovery of rare, biologically relevant cell types or states?

As noted in the Results (page: 12, lines: 443-445), small clusters such as 11 and 13 in the gold-standard dataset were called almost entirely as multiplets by DoubletFinder and DoubletDetection, despite having low cell-hashing multiplet counts. This overcalling illustrates exactly the risk the reviewer points out: false-positive multiplet calls can lead to removal of rare or biologically relevant populations. We therefore added some text in the Discussion (page: 16, lines:75-585) to emphasize that both false negatives and false positives in multiplet detection can mislead downstream analysis.

c) Beyond clustering, what impact does multiplet contamination (and its removal) have on other critical downstream analyses (e.g., differential expression, trajectory inference, cell-cell communication)?”

We agree that multiplet contamination can in principle affect multiple stages of downstream analysis. One of the most basic downstream analyses is differential gene expression (DGE) between clusters, an analysis that can be meaningfully interpreted irrespective of the nature of the data set. So we selected clusters that are strongly affected by multiplet removal and performed DGE. Specifically, we compared two CD14 monocyte clusters before and after multiplet removal (new Figure 11). One of these clusters looses many droplets after multiplet removal and the other is one of the small, disappearing clusters. The pair was chosen to demonstrate how the remaining droplets of the disappearing cluster might display a more representative signal of their cell-type annotation, but this was not the case. While overall log fold-change estimates were highly correlated between conditions (Spearman ρ = 0.96), we observed notable changes in the top differentially expressed genes: several NK cell–associated markers lost statistical significance after multiplet removal, though they remained among the strongest residual signals. There are also quite a few significant DEGs in this analysis and the differentially expressed genes are not exactly the same before vs after multiplet removal (Jaccard = 0.72). We did highlight the canonical markers for CD14 monocytes that were also significant DEGs, and they were not affected by multiplet removal, in contrast to the mentioned NK cell markers. This highlights how multiplets can introduce misleading transcriptional signatures into DGE results and potentially confound biological interpretation. We added a new Methods subsection (page: 7-8, lines: 262-281), a new paragraph in Results (page: 14, lines: 506-530), the new Figure 11, and a small reference in Discussion (page: 16, lines: 580-585) to reflect the findings of this additional analysis.

We did not pursue trajectory inference or cell–cell communication analysis. While these are valuable downstream applications, they are not uniformly applied across scRNA-seq studies, and are only relevant for the respective tissues where differentiation trajectories and/or cell-cell communication is the purpose of the analysis (which was not the case for our data sets). In contrast, multiplet removal, principal component analysis, clustering, cell-type annotation, and DGE are core analyses performed in nearly every scRNA-seq workflow.

We believe these revisions substantially strengthen the manuscript and directly address the reviewer’s concerns.

We thank the editor and the reviewer again for their insightful feedback and hope the revised version will now be suitable for publication.

Sincerely,

Dimitris Ttoouli and Daniel Hoffmann

---

## [Editor Report · Decision Letter 1]

17 Sep 2025

Multiplets in scRNA-seq data: extent of the problem and efficacy of methods for removal

PONE-D-25-32117R1

Dear Dr. Ttoouli,

We’re pleased to inform you that your manuscript has been judged scientifically suitable for publication and will be formally accepted for publication once it meets all outstanding technical requirements.

Kind regards,

Nagarajan Raju

Academic Editor

PLOS ONE

Additional Editor Comments (optional):

Based on the responses to reviewer's comments, we are accepting your article for the publication.
---

## [Editor Report · Acceptance letter]

PONE-D-25-32117R1

PLOS ONE

Dear Dr. Ttoouli,

I'm pleased to inform you that your manuscript has been deemed suitable for publication in PLOS ONE. Congratulations! Your manuscript is now being handed over to our production team.

Kind regards,

on behalf of

Dr. Nagarajan Raju

Academic Editor

PLOS ONE